# Breast-Conserving Surgery Guided with Magnetic Seeds vs. Wires: A Single-Institution Experience

**DOI:** 10.3390/cancers16030566

**Published:** 2024-01-29

**Authors:** Elisa Moreno-Palacios, Covadonga Martí, Laura Frías, Marcos Meléndez, Adolfo Loayza, María José Roca, Vicenta Córdoba, José María Oliver, Alicia Hernández, José Ignacio Sánchez-Méndez

**Affiliations:** 1Obstetrics and Gynecology Department, Breast Unit H Universitario La Paz, 28046 Madrid, Spain; cmartia@gmail.com (C.M.); laura.frias.aldeguer@gmail.com (L.F.); marcosmelendezgispert@gmail.com (M.M.); adolfomusi@gmail.com (A.L.); ahernandezg@salud.madrid.org (A.H.); jisanchezmendez@gmail.com (J.I.S.-M.); 2Radiology Department, Breast Unit H Universitario La Paz, 28046 Madrid, Spain; mariajose.roca@salud.madrid.org (M.J.R.); mvicenta.cordoba@salud.madrid.org (V.C.); josemaria.oliver@salud.madrid.org (J.M.O.)

**Keywords:** breast cancer, non-palpable, breast-conserving surgery, wire, magnetic seed

## Abstract

**Simple Summary:**

Breast conserving surgery (BCS) for non-palpable tumors has witnessed increasing prevalence, primarily attributed to screening mammograms. Currently, there are multiple options to guide resection of non-palpable tumors, with wires being the globally predominant choice due to economic considerations, and the absence of unequivocal superiority of any other alternative. We present our experience with magnetic seeds (Magseed^®^) and compare the use of magnetic seed with wires to guide breast-conserving surgery, finding that the surgical specimen volume in the magnetic seed group was smaller with a statistical significance. This suggests that magnetic seeds facilitate less extensive resections, promising improved cosmetic outcomes while maintaining oncological efficacy. The observed advantages highlight the potential of magnetic seeds as a viable alternative to traditional wire-guided procedures. As further research and trials substantiate these findings, magnetic seeds could become Integral to guiding BCS.

**Abstract:**

Purpose: The aim of this study is to describe our initial experience using magnetic seeds (Magseed^®^) to guide breast-conserving surgery in non-palpable breast lesions and compare the use of magnetic seed with wires to guide breast-conserving surgery in terms of clinical and pathological characteristics. Methods: We performed a retrospective study including all breast-conserving surgeries for non-palpable breast lesions under 16 mm from June 2018 to May 2021. We compared breast-conserving surgeries guided with magnetic seeds (Magseed^®^) to those guided with wires, analyzing tumor and patient characteristics, surgical time, and pathological results of the surgical specimens. Results: Data from 225 cases were collected, including 149 cases guided by magnetic seeds and 76 cases guided by wires. The breast lesion was localized in every case. Both cohorts were similar regarding clinical and pathological characteristics. We found significant statistical differences (*p* < 0.02) in terms of the median volume (cm^3^) of the excised specimen, which was lower (29.3%) in the magnetic seed group compared with the wire group (32.5 [20.5–60.0]/46.0 [20.3–118.7]). We did not find significant differences regarding surgical time (min) or the affected or close margins. Conclusion: In our experience, the use of magnetic seed (Magseed^®^) is a feasible option to guide breast-conserving surgery of non-palpable lesions and enabled us to resect less breast tissue.

## 1. Introduction

In recent years, due to screening programs and the increasing use of neoadjuvant therapy, there has been an increment in breast surgery for non-palpable lesions [1]. Breast-conserving surgery (BCS) is gaining widespread acceptance, as evidence demonstrates that mastectomy provides no survival benefit over BCS plus radiotherapy [2]. The overarching objective of BCS is to excise the targeted tissue with adequate margins, all while preserving healthy breast tissue and achieving an aesthetically acceptable result. The challenge when resecting non-palpable breast tumors lies in what to use to guide resection in the surgery room [3].

Traditionally, wires have been the primary method for guiding the resection of non-palpable breast tumor and remain the most commonly used technique [1,4]. However, wires present several disadvantages. Wires need to be placed shortly before surgery, as they are inserted into the breast tissue with a portion protruding outside of the skin. Therefore, patients must limit their movements to reduce the risk of wire displacement and migration, causing discomfort for the patient and surgical scheduling difficulties. Another limitation is the site of wire insertion, which is normally placed where it is easier for the radiologist and does not consider the surgeon’s cosmetic surgical approach, although the site of insertion can be discussed with the radiologist if close communication is possible [5,6]. 

For surgeons, wires pose a significant disadvantage in the operating room, as they have to mentally plan the surgical approach by visualizing the mammogram with the consequent breast tissue compression, since it is not possible to localize the wire prior to opening the skin. All previously mentioned factors contribute to the difficult localization of non-palpable tumors intraoperatively, potentially resulting in increased breast volume resection. Due to these disadvantages, multiple devices and techniques have been developed and commercialized to guide BCS for non-palpable tumors [3,7,8].

In 2016, the FDA cleared the magnetic seeds Magseed^®^ (MSs) for breast localization, and they have become a feasible alternative to wires [9]. The use of MSs to guide the resection of non-palpable breast lesions presents many advantages: precise surgical guiding, less discomfort for patients, and the facilitation of surgical scheduling, as the seed can be placed days before surgery [10].

The aim of this study is to describe our initial experience using MSs to guide BCS for non-palpable breast lesions, as well as to compare our experience using MSs with wire-guided BCS in terms of clinical and pathological characteristics.

## 2. Materials and Methods

We performed a single-institution retrospective study including every patient that underwent BCS for non-palpable tumors under 16 mm from June 2018 to May 2021. The first MS was placed in January 2019, and for several months, we used either wires or MSs to guide breast surgery for non-palpable breast lesions depending on the availability of MSs, until wires were completely substituted by MSs. During this period, no other devices were used for guiding BCS. Every patient signed an informed consent form prior to the insertion of either the wire or the MS.

The inclusion criteria for this study were defined as follows: individuals who underwent wire-guided or MS-guided BCS for non-palpable breast tumor radiologically measuring 16 mm or less at the time of surgery. The data of each patient were obtained by retrospectively reviewing every medical record.

Each patient underwent a mammogram and an ultrasound at diagnosis. Magnetic resonance was performed based on the radiologist’s criteria if necessary to complete the study. Upfront treatment was decided upon the multidisciplinary committee according to international guidelines [11]. Both wires and MSs were placed by four experienced breast radiologists under ultrasound guidance, as stereo placement was not available at the time of the study. MSs were placed using local anesthesia. A control mammogram was performed after placing the wire or the MSs in all cases and could be visualized during the procedure in the operating room (Figure 1). Wires were placed on the same day of the surgery; each patient first went to the radiological room, had the wired placed, and then was taken to the surgery room or to their hospital room until the theater was available. MSs were placed independently of the surgery day, always less than 30 days before surgery, as MSs were initially approved for placement up to 30 days prior surgery (although currently, MSs have no time limit between placement and surgery). Every surgery was performed by one of the six experienced breast surgeons of the breast unit, and an informed consent form was signed before surgery [3]. MSs were intraoperatively localized using the Sentimag^®^ probe. Every surgical specimen, including wire- and MS-guided specimens, was evaluated with a mammogram to confirm adequate excision of the lesions (Figure 2). If the mammogram of the surgical specimen showed close margins, the surgeon was informed, and the margin was extended in the same surgery.

Data were retrospectively collected, including age, body mass index (BMI), menopausal status, histological type, grade, hormone receptor status, HER2 status, ki67, tumor phenotype (St Gallen 2013), surgical guidance technique, primary treatment (surgery or neoadjuvant treatment, either chemotherapy or hormonotherapy), radiological tumor size, surgical time, total volume of the surgical specimen (specified in the pathological report, adding the volume of the surgical specimen of the extended margins if performed), margin status, and possible re-excision. Surgical specimen margins were categorized into three groups: free, close margins (<2 mm), and ink touch group.

We compared BCSs guided with MSs with those guided with wires performed from June 2018 to May 2021. A descriptive analysis of the collected data was conducted, with categorical variables described as frequency or percentage. Continuous variables were reported as the median and interquartile range (IQR). A comparison of characteristics among variables was performed using the most appropriate test. Since the majority of variables had a non-normal distribution, the test commonly used was the Mann–Whitney test. The type 1 error rate (ⲁ) was set to 0.05. The confidence interval was set to 95%. To compare the surgical specimen volume, we employed the Mann–Whitney test. Statistical analyses were performed using SPSS v26 (IBM Corporation, Armonk, NY, USA). We used the STROBE cohort reporting guidelines. Ethical approval was obtained previously (PI-5437).

## 3. Results

A total of 225 patients that underwent BCS for non-palpable breast lesions were included in the study; 149 surgeries were MS-guided, and 76 were wire-guided. No incidence during the placement of MSs or wires was reported, being accurately placed in all cases either centrally or laterally to the lesion. The tumor was localized and excised in every case. In six cases, two MSs were placed to guide excision: in four cases, two coils were placed prior to neoadjuvant treatment for marking purposes, and in two patients, they were placed to delimit an area of microcalcifications with confirmed diagnosis of intraductal carcinoma (Figure 3). Both cohorts exhibited similarity in terms of clinical and pathological characteristics. In the MS group, the median age was 65 years (range 28 to 85), while in the wire group, it was 60 years (range 39–90), with no statistical difference found (*p* = 0.60). 

Postmenopausal status was observed in 69.9% of the patients in the MS group and 76% of patients in the wire group. The median BMI was 25.2 in the MS group and 24.5 in the wire group, with no statistically significant differences (*p* = 0.32).

The majority of tumors in both groups were ductal carcinomas, and no statistically significant difference was found in terms of the histological grade. The tumor phenotypes in the MS group were as follows: 62 cases were Luminal A, 27 were Luminal B, 16 were Luminal B HER2, 12 were HER2, 15 were basal-like, and 16 were intraductal carcinomas. In the wire group, the tumor phenotypes were distributed as follows: 11 intraductal carcinomas, 21 Luminal A, 16 Luminal B, 11 Luminal B HER2, 7 HER2, and 9 basal-like.

The median radiological tumor size was 8 mm, ranging from 0 to 16 mm, with no statistically significant difference between the two groups (*p* = 0.64). Initial treatment was surgery in 82 patients in the MS group and 35 in the wire group. Neoadjuvant endocrine therapy was administered to 27 patients in the MS group and 9 patients in the wire group, while neoadjuvant chemotherapy was given to 40 and 32 patients in the MS group and the wire group, respectively (Table 1).

The median radiological size before surgery of the breast lesions was 8 mm (range 0–16 mm), with no statistical difference between the MS and the wire groups. In 57 cases (35 MSs and 22 wires), no residual tumor was localized radiologically after neoadjuvant chemotherapy, and consequently, the MS or the wire was placed in the marking coils for guiding excision of the tumor bed.

The median volume of the surgical specimen was 32.5 cm^3^ (20.5–60.0) in the MS group and 46 cm^3^ (20.3–118.7) in the wire group, with a statistically significant difference (*p* < 0.02). The median surgical time was 92 min (range 45–200) and 100 min (range 40–260) in the MS and the wire group, respectively (*p* = 0.17). Several patients underwent unilateral or bilateral oncoplastic procedures after the surgical removal of the tumor, consequently extending surgical time. 

Free margins were obtained in 91% (n = 136) of cases in the MS group and 94% (n = 72) of cases in the wire group. A total of 17 patients underwent re-excision (7.5%), including 4 cases in the wire group (5.3%) for close margins, and 13 cases in the MS group (8.7%), 9 for close margins, and 4 for ink touch. When comparing close margins, no statistical difference was found between both groups (*p* = 0.34) (Table 1).

## 4. Discussion

MSs have demonstrated their feasibility and non-inferiority compared to wires for guiding BCS in non-palpable breast tumors [12,13,14]. Furthermore, the utilization of MSs has been proven to offer several distinct advantages over traditional wire-guided procedures. One of the key advantages of MSs is the possibility of dissociating placement and surgical timing, thereby facilitating surgical scheduling [15]. Our institution completely ceased using wires and switched to using MSs, but in the beginning, the availability of MSs was limited, and thus we had to use either wires or MSs based solely on availability. Consequently, this unique period allowed us to compare both methods of intraoperative localization. 

During this period, we found that independent scheduling by the radiologist and surgeons significantly enhanced the efficiency of surgical scheduling. The practice of placing wires prior to surgery often led to delays in surgical timing, serving as one of the primary reasons for substituting wires with MSs. 

We successfully reported the placement of the MS in the lesion, either centrally or laterally, as well as the detection of the MSs and excision of the lesion in every case. No complications were registered during MS placement or excision. Additionally, we utilized MSs for bracketing breast lesions, understood as bracketing the placement of two or more MSs to delimitate the boundaries of a tumor lesion that we want to excise [16]. The bracketing technique was first described with wires, but has been reported with other devices such as MSs and radioactive seeds [17,18]. We report six cases in which two MSs were placed to guide excision: four cases in which two coils were placed prior to neoadjuvant treatment for marking purposes, and in two patients, they were placed to delimit an area of microcalcifications with a confirmed diagnosis of intraductal carcinoma. In every case, both magnetic seeds were excised in the surgical specimen. 

We did not collect data on the learning curve, but our radiologist became familiarized with the device in a short period of time, in agreement with findings in other studies [15,19]. In conclusion, it can be stated that the MS technique involves a short learning curve for radiologists and surgeons. However, it should be noted that our multidisciplinary breast unit diagnoses and treats over 360 breast cancer cases per year. Therefore, the learning curve may not be reproducible in every unit and may depend on the experience of radiologists and breast surgeons. 

In our study, we found smaller surgical specimens in the MS group compared to the wire group in terms of volume. To our knowledge, only a limited number of studies have found differences in the specimen size when comparing guiding techniques. Micha et al. found that the surgical specimen’s weight was smaller in the BCS group guided with MSs than with wires, showing a statistically significant difference [20]. Similarly, Redfern and Shermis reported significantly lower total volumes resected in the MS group compared with the wire-guided group [21]. Conversely, the iBRA-NET group did not find a difference between groups [22]. It is important to take into account that they included T1–T3 tumors, introducing potential selection bias as the localization modality used depended on the surgeon’s preference. 

In our study, we measured volume (cm^3^) instead of weight, finding smaller surgical specimens when MSs were used for guidance compared to wires, with this difference being statistically significant. In our opinion, MSs enable us to target the breast tissue for resection more precisely by localizing them with the Sentimag probe rather than relying on the tip of the wire. This is particularly important considering the risk of displacement. It has been demonstrated that when resecting breast tissue, excising smaller volumes leads to better cosmetics results, as the breast defect is reduced [23,24]. Although a recent review concluded that there is no evidence that MSs reduce surgical specimen volume, our findings, along with further studies, may challenge this assumption in the near future [12].

Cosmetic outcomes are not solely determined by the volume of the surgical specimen but are also influenced negatively by potential re-excision procedures and an increase in scar length [25]. Re-excision rates for MSs have been reported in the literature, ranging from 5.1 to 21.9% [26,27]. Gera et al. published a re-excision rate of 11.25% in a pooled analysis that included 1559 MS procedures [12]. In our study, we observe a low re-excision percentage, although no statistical difference was found between MSs and wires. Additionally, despite resecting a smaller breast volume in the MS group, we did not observe an increase in close and/or positive margins, leading to no additional re-excision procedures in this group. 

Another significant advantage we identified when we started to use MSs was the possibility of the localization of the seed with the Sentimag^®^ probe before skin incision. This capability allows for the design of the surgical approach, reducing breast dissection and scar length, ultimately contributing to obtaining better cosmetic results [24]. It is plausible that the ability to design the surgical approach before making the incision contributes to reducing the surgical specimen volume as well. 

Surgical time varied widely, ranging from 90 to 200 min in the MS group, and 90 to 260 min in the wires group, respectively, with no statistical differences found, consistent with the findings in many studies [28]. This variability can be attributed to some patients undergoing unilateral or bilateral oncoplastic surgeries to ensure good cosmetic results, even at the expense of extending surgical time. Additionally, bracketing procedures can be more time-consuming, contributing to the observed range [29]. 

The primary drawback of MSs is their cost, as they are significantly more expensive than wires. However, it is important to note that the overall costs can potentially be reduced through more efficient surgical and radiological timing. Further research in cost-efficiency is needed [30].

The main limitation of our study is that it is a single-center, small-sample, retrospective observational study. On the other hand, a notable strength of our study lies in the fact that the surgeons’ preference did not introduce bias. None of the surgeons participated in the decision-making process regarding which device to place; MSs were only utilized if they were available at that moment. 

In summary, our primary objective when performing BCS for breast cancer is to resect the lesion with free margins and achieve an acceptable cosmetic result. Numerous factors can influence cosmetic outcomes, not only surgical factors but also adjuvant treatments, especially radiotherapy. Evaluating cosmesis results after breast cancer treatment is challenging, as no universally accepted evaluation system exists, and there is currently no comparable scoring system. Moreover, it appears that surgeons’ perceptions of cosmesis outcomes do not align with those of patients, with patients generally expressing higher satisfaction with BCS results than surgeons [31]. While we did not directly assess cosmetic results, there is clear evidence that reducing the amount of breast tissue excised leads to better aesthetic results. In our experience, MSs allow for less extensive breast resection. 

While acknowledging the promising results observed in our study, it is crucial to highlight the need for further research and exploration in this area. Multi-center prospective studies, ideally incorporating randomized trials, will be instrumental in corroborating and expanding upon our findings. The ongoing investigation will contribute to a more comprehensive understanding of the benefits and implications of magnetic-seed-guided BCS, providing valuable insights for future advancements in breast cancer surgical techniques.

## 5. Conclusions

In conclusion, based on our experience, the use of magnetic seeds (Magseed^®^) enables us to perform less extensive resections in BCS, contributing to obtaining better cosmetic outcomes while maintaining comparable oncological results. Additionally, we have observed that magnetic seeds facilitate hospital logistical organization by streamlining the cooperation between radiology and surgery departments. The seamless integration of magnetic seeds facilitates efficient coordination between radiology and surgery departments. This synchronization proves instrumental in optimizing scheduling, reducing delays, and enhancing overall procedural efficiency. The practical advantages extend beyond the operating room, impacting the broader organizational aspects of patient care.

## Figures and Tables

**Figure 1 cancers-16-00566-f001:**
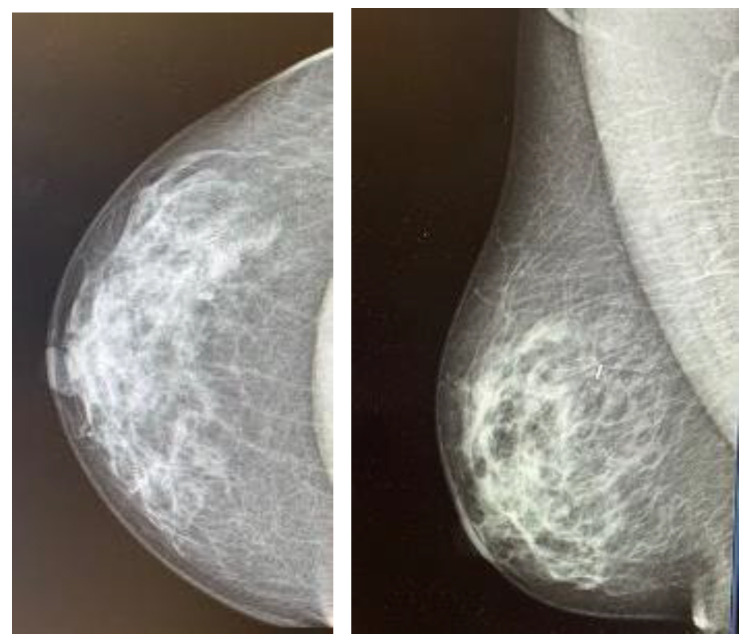
Control mammogram performed after placement of the magnetic seed.

**Figure 2 cancers-16-00566-f002:**
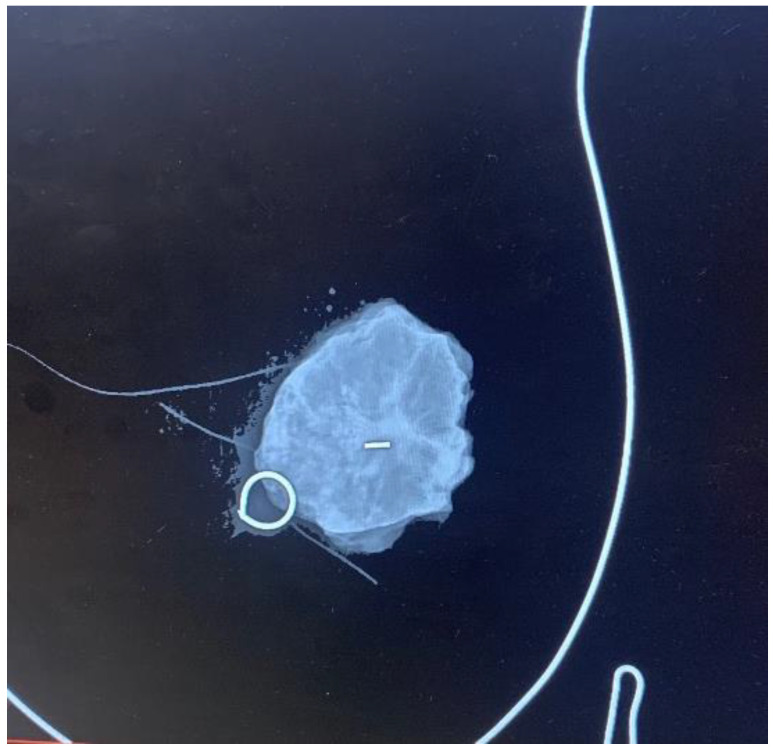
Control mammogram of the surgical specimen. The surgical specimen is positioned on a methacrylate plate featuring a breast drawing to orient it anatomically.

**Figure 3 cancers-16-00566-f003:**
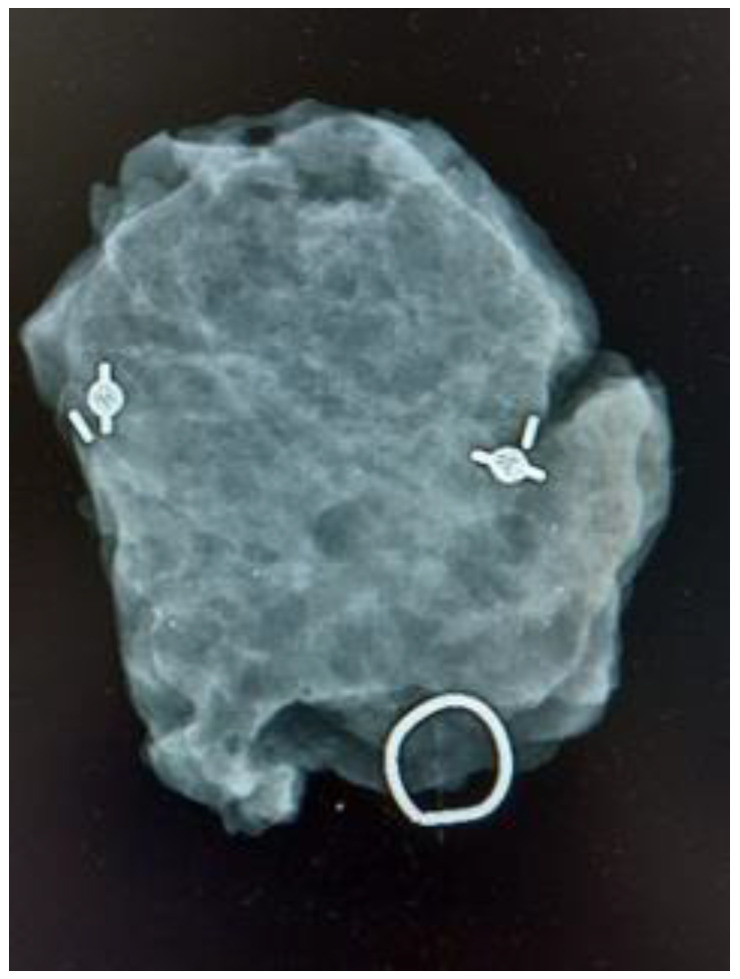
Control mammogram of a surgical specimen guided with two magnetic seeds.

**Table 1 cancers-16-00566-t001:** Patient and tumor characteristics. ^1^ Magnetic seed (Magseed^®^).

		MS ^1^		Wire		*p*
		n	%	n	%	
		149	100	76	100	
Age (years)	median (range)	65.0 (28–85)		60.0 (39–90)		0.60
Menopausal Status	premenopausal	45	30.2	18	23.7	0.30
	postmenopausal	104	69.8	58	76.3	
BMI	median (range)	24.5 (17.72–39.66)		25.2 (16.85–41.65)		0.32
Histological Type	Ductal	107	71.8	55	72.4	0.75
	Lobular	12	10.7	5	14.5	
	In situ	16	8.1	11	6.6	
	Others	14	9.4	5	6.6	
Differentiation Grade	G1	37	24.8	16	21.1	0.53
	G2	66	44.3	38	50.0	
	G3	43	28.9	22	28.9	
	unknown	3	2.0	0	0.0	
Tumor Phenotype	Luminal A	62	46.9	21	32.8	0.49
	Luminal B	27	20.5	16	25.0	
	Luminal B Her2	16	12.1	11	17.2	
	Her2	12	9.1	7	10.9	
	Basal	15	11.4	9	14.1	
Tumor Size per image (mm)	median (range)	8 (0–16)		8 (0–16)		0.64
Focality	Unifocal	138	92.6	66	86.8	0.16
	Multifocal-Multicentric	11	7.4	10	13.2	
Initial Treatment	Excision	82	55.0	35	46.1	0.06
	neoadjuvant endocrine therapy	27	18.2	9	11.8	
	neoadjuvant chemotherapy	40	26.8	32	42.1	
Surgical Time (min)	median (range)	92 (45–200)		100 (40–260)		0.17
Volume of the excised specimen (cm^3^)	median (range)	32.5 (20.5–60.0)		46,0 (20.3–118.7)		0.02
Margins	Free	136	91.3	72	94.7	0.34
	Close (<2 mm)	9	6.0	4	5.3	
	Touch ink	4	2.7	0	0.0	

## Data Availability

The datasets used and analyzed during the current study are available from the corresponding author on reasonable request.

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
