# Peer review of "Breast-Conserving Surgery Guided with Magnetic Seeds vs. Wires: A Single-Institution Experience"

_cancers, 2024, doi:10.3390/cancers16030566_

Round 1
Reviewer 1 Report
Comments and Suggestions for Authors
change all nonpalpable to non-palpable.
Line 22 change enable to enabled.
]line 31 - remove the word exceeding
line 34 remove the words and till the moment and change to "until now"
I will stop now with the English changes, this paper needs editing for correct use of english.
To cpmare MS and wires we need a comparison of radiological size betwen the groups, it would be good to know the radiological size of the invasive tumours on average in each group and and the radioloigcal size of the preinvasive. ie were the tumours to be resected bigger in one of the groups. also need to know how many were stereo placed in each group and how many US placed as stereo placement may lead to laerger volumes being removed.
For assessment of volume - can only use those having single site resections - so unifocal disease, probably best to compare unifocal disease T1-T2, so you are comparing like for like and excluding multifocal cancers and large areas of dcis. also need to exclude any that had oncoplastuc resections for large volumes eg mammoplasty. not sure whether you should include NACT in the size of the lesions for comparison, depends on the resection volumes vs size of index lesion.
When comparing the numbers in table 1 you need to compare the rates not the individual numbers, eg rate of dcis in MS vs rate in wire group. the rate of dcis is higher in the wire group compared to the invasive cancers (although not significant.
I think the analysis needs to be redone for T-T2 invasive cancers only so there is direct comparison , size of dcis needs to be clear, if there is no difference in the size of dcis lesions betwen the two groups then can also compare volume for these lesions. many confounders here that need to be accounted for. size of the original radiological lesion being the main confounder. please coompare this between groups and volume resected is dependant on this radiological size.
currently there is the potential for confounders and bias for influencing the results, once reanalysed if there is a difference then it may be a true difference, currently not clear whether the groups are significantly different.
Add multifocal tumour number and bracketing numbers to Table 1.
line 35 change laces to placed.
Comments on the Quality of English LanguageEnglish needs improvement, understandable but quality needs improving.
Author Response
- change all nonpalpable to non-palpable.
ANSWER: Thank you for your comment, I have revised and change it in the manuscript.
- Line 22 change enable to enabled.
ANSWER: It has been changed in the manuscript.
- line 31 - remove the word exceeding.
ANSWER: It has been changed in the manuscript.
- line 34 remove the words and till the moment and change to "until now."
ANSWER: It has been changed in the manuscript.
- I will stop now with the English changes, this paper needs editing for correct use of English.
ANSWER: Thank you very much for your comments. A full-length English editing has been completed.
- To compare MS and wires we need a comparison of radiological size between the groups,
ANSWER: Thank you very much for the comment, in fact, you are completely right. We conducted a comparison of the radiological size in each group and found no statistical difference. The detailed data has been incorporated into the manuscript and is available in Table 1.
- It would be good to know the radiological size of the invasive tumors on average in each group.
ANSWER: That is an interesting comment. We conducted a comparison of both groups, and our findings reveal no statistically significant difference in the size of invasive tumors between the two groups (p=0.382).
- and the radiological size of the preinvasive. ie were the tumors to be resected bigger in one of the groups.
ANSWER: Dear reviewer, thank you very much for this insightful comment. Upon analyzing the data, we observed that lesions guided with wires were slightly larger than those with magnetic seeds, although the difference was not statistically significant (p=0.610).
- also need to know how many were stereo placed in each group and how many US placed as stereo placement may lead to larger volumes being removed.
ANSWER: Thank you for your comment. We can confirme that every wire and magnetic seed was placed using ultrasound as at that time we did not have the necessary needle for stereotactic insertion. It has been modified in the manuscript.
- For assessment of volume - can only use those having single site resections - so unifocal disease, probably best to compare unifocal disease T1-T2, so you are comparing like for like and excluding multifocal cancers and large areas of dcis. also need to exclude any that had oncoplastic resections for large volumes eg mammoplasty.
ANSWER: That is a very interesting comment. Indeed, the median radiological size of the tumors in both groups was 8mm, range 0-16mm. We specifically included tumors with a size of 16mm or less (T1). Among these, only 2 in situ tumors were found to be multifocal. There was a total of 198 unifocal tumors. Importantly, no statistical difference was observed (p=0.461).
- not sure whether you should include NACT in the size of the lesions for comparison, depends on the resection volumes vs size of index lesion.
ANSWER: Thank you for your comment. We incorporated cases showing a concentric response to chemotherapy, as assessed by magnetic resonance imaging, wherein the radiological size of the remaining breast lesion after neoadjuvant chemotherapy (NACT) was limited to 16mm or less. Our surgical approach is tailored to the remaining tissue post-chemotherapy, with no intention to excise the entire tissue bed. This specific subgroup is particularly intriguing and represents a growing trend in current practices. The rationale behind administering neoadjuvant chemotherapy is to effectively reduce tumor size, thereby minimizing the extent of surgery. Our surgical procedures align with this premise.
- When comparing the numbers in table 1 you need to compare the rates not the individual numbers, eg rate of dcis in MS vs rate in wire group. the rate of dcis is higher in the wire group compared to the invasive cancers (although not significant.
ANSWER: Thank you for your intriguing comment. The statistical method employed for data analysis has already considered this aspect. However, we appreciate your suggestion, and in response, we have modified Table 1 to include this relevant data.
- I think the analysis needs to be redone for T-T2 invasive cancers only so there is direct comparison , size of dcis needs to be clear, if there is no difference in the size of dcis lesions between the two groups then can also compare volume for these lesions. many confounders here that need to be accounted for. size of the original radiological lesion being the main confounder. please compare this between groups and volume resected is dependent on this radiological size.
currently there is the potential for confounders and bias for influencing the results, once reanalyzed if there is a difference then it may be a true difference, currently not clear whether the groups are significantly different.
ANSWER: Thank you very much for your observations. There is no statistical difference in size between the infiltrating carcinomas and the in-situ carcinomas (p=0.106).
There is no statistical difference when we analyze each group individually. Even without excluding the radiological complete response or size 0mm, that should be excluded.
- Add multifocal tumor number and bracketing numbers to Table 1.
ANSWER: It has been changed in the manuscript.
- line 35 change laces to placed.
ANSWER: It has been changed in the manuscript.

Reviewer 2 Report
Comments and Suggestions for Authors
The study described initial experience using magnetic seeds (Magseed® ) to guide breast-conserving surgery in nonpalpable breast lesions and compared the use of magnetic seed with wires to guide breast-conserving surgery. The following are questions and concerns that the authors need to address.
- Material and methods:
1. Please elaborate in more detail on the inclusion and exclusion criteria. Where is the data source come from? Please add in.
2. From lines 73 to 84, please explain why the authors followed those steps. Please add references.
3. Please demonstrate more detail in the statistical analysis. How about confidence intervals (90% CIs or 95% CIs, etc)?
-Which test was used to investigate the association? Please add in.
- Results:
In table 1: Why the authors report median value instead of mean value. Please explain and add whether data is normally distributed or not in the statistical analysis.
Comments on the Quality of English Language
It would be better to have an English scientific write to review and smooth the whole article.
Author Response
The study described initial experience using magnetic seeds (Magseed® ) to guide breast-conserving surgery in nonpalpable breast lesions and compared the use of magnetic seed with wires to guide breast-conserving surgery. The following are questions and concerns that the authors need to address.
- Material and methods:
- Please elaborate in more detail on the inclusion and exclusion criteria. Where is the data source come from? Please add in.
ANSWER: Thank you very much for your comment. The information has been included in the manuscript.
- From lines 73 to 84, please explain why the authors followed those steps. Please add references.
ANSWER: Thank you for your comment.
We explain the standard protocol in our unit for wire and magnetic seed insertion. This protocol is in accordance with the description of the techniques (Kapoor MM et al. (3)).Wires were placed on the day of the surgery, as they are inserted in the breast tissue with part of it protruding outside of the skin. As a result, patients were required to restrict their movements to minimize the risk of wire displacement and migration, which could cause discomfort. It was deemed crucial to minimize the time between wire insertion and surgery to address these challenges.
Magnetic seeds (MS) were placed independently of the surgery day, always less than 30 days before surgery, as MS were initially approved for placement up to 30 days prior surgery. Although currently MS has no limit time between placement and surgery.
Magseeds (MS) were intraoperatively localized using the Sentimag® probe. Each surgical specimen, including those guided by both wires and MS, underwent evaluation with mammography to ensure the proper excision of the lesions and either the MS or the entire length of the wire (see Figure 2). This process aligns with the technique described by Lamb et al. (13). If the mammogram of the surgical specimen indicated close margins of the tumor or microcalcifications, the surgeon was promptly informed, and the margin was extended during the same surgical procedure.
- Please demonstrate more detail in the statistical analysis. How about confidence intervals (90% CIs or 95% CIs, etc)?
ANSWER: This is a very interesting comment. The Confidence Intervals for the statistical tests were set at 95%. This information has been incorporated into the manuscript.
-Which test was used to investigate the association? Please add in.
ANSWER: Thank you for your comment. In each case, the most suitable test was employed. Since the majority of variables had a non-normal distribution, the test commonly used was the Mann-Whitney test: a non-parametric test for comparing continuous and nominal variables. This clarification has been added to the manuscript.
- Results:
In table 1: Why the authors report median value instead of mean value. Please explain and add whether data is normally distributed or not in the statistical analysis.
ANSWER: Thank you for your comment. Due to the non-normal distribution of the majority of variables, we opted to calculate the median value instead of the mean.
-It would be better to have an English scientific write to review and smooth the whole article.
ANSWER: Thank you very much for your comments. A full-length English editing has been completed.

Reviewer 3 Report
Comments and Suggestions for Authors
The authors present the data of an interesting retrospective study on the comparison between two different techniques for marking small non-palpable breast tumors. The text is well-written and easy to read, with sufficient explanations in the methods section to ensure reproductibility. Strenghts and limitations are mentioned in the discussion section. The authors show that Magnetic Seeds are a viable and safe alternative to wires, presenting also several advantages.
Comments on the Quality of English LanguageSome minor corrections are required, mostly a revision of misspelling and some grammar adjustments.
Author Response
REVISOR 3
-The authors present the data of an interesting retrospective study on the comparison between two different techniques for marking small non-palpable breast tumors. The text is well-written and easy to read, with sufficient explanations in the methods section to ensure reproducibility. Strengths and limitations are mentioned in the discussion section. The authors show that Magnetic Seeds are a viable and safe alternative to wires, presenting also several advantages.
Some minor corrections are required, mostly a revision of misspelling and some grammar adjustments.
ANSWER: Thank you very much for your comment. A full-length English editing has been completed.

Round 2
Reviewer 1 Report
Comments and Suggestions for Authors
There is a large study comparing wires vs magseed recently published by the Ibra-net group. Would be worth in the discussion discussing why the resection volumes in the wire group are much larger than in the Magseed group in your study. Would be good to mention the limitations of your study - single centre, small sample size and to discuss in your opinion why there is a smaller volume in the MS arm. Is there anything specific about practice in this unit that can give us the answer.
in the methods please tell us how you did the comparison - eg analysis comparing resection volume was done on unifocal invasive cancers undergoing simple wide local excision.
In the results section can you analyse re-excision rates between the two groups please. e.g 13/149 vs 4/76 -ns but worth putting in .
in the discusion it says in conclusion there is a short learning curve for radiologists and surgeons - there is no data reported in the results on learning curves. suggest changing.
The strength of our study lies in is the finding of the differences 246 inon the surgical specimen volume. - this is not a strength of the study. this is merely a result. a limitation of the study is the nature of the study and the small sample size. Probably better to say that in our study we found smaller volume, this is in line with other studies such as x,y and z but the ibranet study with a larger sample size found no difference. and discuss why your study may have found a difference and whether you think it is a real difference and why.
also is breast size a measured outcome ? ie did patients in the wire arm have a larger breast and larger BMI? if not measured this is a possible confounder.
Comments on the Quality of English Language
needs minor improvement still
Author Response
-There is a large study comparing wires vs magseed recently published by the Ibra-net group. Would be worth in the discussion discussing why the resection volumes in the wire group are much larger than in the Magseed group in your study.
ANSWER: Thank you for this insightful comment. We have incorporated a discussion of this study into our research.
-Would be good to mention the limitations of your study - single center, small sample size
ANSWER: Thank you for your comment. We have modified the manuscript.
-to discuss in your opinion why there is a smaller volume in the MS arm. Is there anything specific about practice in this unit that can give us the answer.
ANSWER: Thank you for your comment. In our opinion, MS enables us to target the breast tissue for resection more precisely by localizing it with the Sentimag probe rather than relying on the tip of the wire. This is particularly important considering the risk of displacement. We have added this information to the manuscript.
-in the methods please tell us how you did the comparison - eg analysis comparing resection volume was done on unifocal invasive cancers undergoing simple wide local excision.
ANSWER: This information has been added to the text.
-In the results section can you analyze re-excision rates between the two groups please. e.g 13/149 vs 4/76 -ns but worth putting in .
ANSWER: Thank you for your comment. We have added to the result section.
-in the discussion it says in conclusion there is a short learning curve for radiologists and surgeons - there is no data reported in the results on learning curves. suggest changing.
ANSWER: Thank you for your comment. In fact, you are totally correct, we have not collected data on the learning curve. We have modified it in the manuscript.
-The strength of our study lies in is the finding of the differences 246 inon the surgical specimen volume. - this is not a strength of the study. this is merely a result. a limitation of the study is the nature of the study and the small sample size. Probably better to say that in our study we found smaller volume, this is in line with other studies such as x,y and z but the ibranet study with a larger sample size found no difference. and discuss why your study may have found a difference and whether you think it is a real difference and why.
ANSWER: Thank you for your comment. We have modified the manuscript considering your comment.
- also is breast size a measured outcome ? ie did patients in the wire arm have a larger breast and larger BMI? if not measured this is a possible confounder.
ANSWER: This is a very interesting comment. In fact, you are totally right, breast size could be a confounder. Unfortunately, breast size was not measured. But there was no statistical difference between both groups regarding BMI, (24.5 vs 25.2, p= 0.32)
